# Assessing the Efficiency of a Drinking Water Treatment Plant Using Statistical Methods and Quality Indices

**DOI:** 10.3390/toxics11120988

**Published:** 2023-12-05

**Authors:** Alina Bărbulescu, Lucica Barbeș

**Affiliations:** 1Department of Civil Engineering, Transilvania University of Brașov, 5 Turnului Str., 500152 Braşov, Romania; alina.barbulescu@unitbv.ro; 2Department of Chemistry and Chemical Engineering, Ovidius University of Constanța, 124 Mamaia Bd., 900112 Constanţa, Romania; 3Doctoral School of Biotechnical Systems Engineering, Politehnica University of Bucharest, 313, Splaiul Independenţei, 060042 Bucharest, Romania

**Keywords:** water quality, treatment plant, water parameters, efficiency indices

## Abstract

This study presents the efficiency of a drinking water treatment plant from Constanța, Romania. Individual and aggregated indices are proposed and built using nine water parameters for this aim. The analysis of individual indices permits the detection of the period of malfunctioning of the water treatment plant with respect to various parameters at various sampling points. In contrast, the cumulated indices indicate the overall performance of the treatment plant during the study period, considering all water parameters. It was shown that the outliers significantly impact the values of some indices. Comparisons between the simple average and weighted average indices (built taking into account the importance of each parameter) better reflect the impact on the water quality of some chemical elements that might harm people’s health when improperly removed.

## 1. Introduction

Water is an essential resource for life. Human history shows that the primary freshwater sources have been rivers. They still play a significant role in socio-economic development [1]. In the last decades, water quality has been affected by environmental pollution produced by anthropic activities, becoming inappropriate for drinking, irrigation, and other uses [2]. Therefore, its consumption can harm organisms, especially humans, given that more than two-thirds of organisms are formed of water [3].

Unfortunately, people in some regions or countries lack sufficient access to clean water or use water from contaminated sources with disease-carrying organisms, pathogens, or unacceptable levels of toxic substances and suspended solids [4,5,6]. Olukanni et al. [7] show that over 2.2 million people in developing countries die annually from diseases provoked by contaminated water. Inefficient water treatment and the distribution of drinking water, as well as the consumption of contaminated water, can lead to the apparition of many diseases [8]. To avoid such effects, drinking water must be tasteless, odorless, and colorless, and free from physical, chemical, and biological contaminants.

An extended analysis of the factors affecting the spatial variation in stream water composition is presented in [9], emphasizing natural causes. The surface water quality and the pollutants’ transport can be assessed utilizing statistical methods [10,11,12,13,14] and water quality indicators [15,16,17]. Modeling and forecasting water quality and the parameters that influence it has been performed recently by Artificial Intelligence methods (Fuzzy techniques, ANFIS, C&RT) and hybrid method [14,18,19,20]. Water quality simulation and forecast utilizing exponential models, differential equations, deep learning neural networks, and fuzzy clustering have been developed by some scientists [21,22,23,24].

Romania has abundant sources of drinking water. However, the demand for water resources is constantly rising due to population growth, intensified agricultural and industrial activities, and the recent years of low rainfall and adverse conditions, which impact the quality of drinking water sources [10]. The quality of drinking water is essential for EU residents [25]. The necessary treatments for producing drinking water, depending on the quality of water sources, are presented in Directive EC 2184/2020 [26]. Researchers’ studies reflect the interest in the topic [27,28,29,30,31,32]. Romania is also tasked with finding cost-effective and innovative approaches that address environmental, regulatory, and public concerns for maintaining a clean environment [33,34].

Since ensuring good water quality is essential for the population’s health, research has been developed to propose advanced technologies for drinking water treatment. Some of the most recent technologies are presented in the books and articles of Thomas and Burgess [35], Brar et al. [36], Vara Prasad [37], Caratar et al. [38], Brusseau et al. [39], and Farhaoui and Derraz [40].

Most studies written by Romanian scientists present wastewater analysis, proposing solutions for cleaning them [41,42,43,44,45]. Chirilă et al. [31] studied the water supply sources in Constanta town (Romania), the applied treatments based on their quality, and the performances of the water purification process. Some authors [46,47,48] addressed the disinfection by-products in drinking water, modeling the chlorine decay or proposing the analysis of the chlorine concentration in the distribution system.

This study aims to fill a gap in the knowledge related to the efficiency evaluation of a drinking water treatment plant utilizing a series of individual and aggregated indices introduced by the authors. The originality of this work consists of (1) proposing individual and composite efficiency indices for assessing the plant’s efficiency, (2) building indices that are not restricted to a certain number of parameters or a determined period, and (3) introducing an objective evaluation method of the treatment plant’s efficiency.

## 2. Materials and Methods

### 2.1. Studied Region and Data Series

Constanța city is situated in the Dobrogea region, in the southeastern part of Romania (Europe), with one of the biggest metropolitan areas in Romania. It has a circular drinking water distribution system with a length of about 575 km. The hydrographic region of Dobrogea contains two river basins: the Littoral basin and a portion of the Danube basin (341.5 km along the Danube River), covering an area of 11,809 km^2^ (excluding the Danube Delta), with a network length of 1624 km and an average density of 0.13 km/km^2^. Approximately 73% of this hydrographic network is affected by drying phenomena.

To ensure the best quality and circulation of drinking water, four treatment–storage and pumping stations operate—Constanța Nord, Constanța Sud, Călăraşi, and Palas complex. Groundwater and surface water sources are used for the city’s water supply. The groundwater sources include Caragea Dermen, Cișmea I A, B, C, Cișmea II, Constanța Nord, and Medgidia. The surface water is extracted from the Priza Galeșu (44°15′0″ N and 28°25′60″ E) located on the Danube–Poarta Albă–Midia Năvodari channel. The Caragea Dermen source, situated between Constanța and Ovidiu, is the oldest groundwater source, consisting of 18 wells with depths from 35 to 90 m. It provides water to Ovidiu, Mihail Kogălniceanu, as well as the Palazu Mare neighborhood and the Călărași storage–pumping complex, with a flow rate of 3549 m^3^/h. Cișmea I A, B, and C consist of three groups of 36 wells located in the northern part of Constanța, with a total captured flow rate of about 8500 m^3^/h. The Cișmea I sources provide water to neighborhoods in the northern part of Constanța (also pumped to the Palas storage–pumping complex and the Călărași complex). Cișmea II is situated between the Caragea Dermen and Cișmea I sources and contains ten wells with a captured flow rate of approximately 1700 m^3^/h. The water from this source is transported to the Palas storage–pumping complex. The Constanța Nord source, situated in the northern part of Constanța, south of the Siutghiol Lake, consists of 5 wells with a captured flow rate of about 2200 m^3^/h. The water is pumped from here to the Constanța Nord complex. The Medgidia source is located along the Danube–Black Sea Canal and comprises 11 wells with a captured flow rate of approximately 1500 m^3^/h. The extracted water is pumped to the Constanța Sud complex. The surface source Galeșu captures water from the Poarta Albă–Midia Năvodari Channel at km 6 + 398 and pumps it to the Palas Constanța storage–treatment and pumping complex at 17.4 km. It provides a water supply of 13,050 m^3^/h. This surface source was created to meet the high summer water demand and to supplement the water supply for Constanța city if necessary. The intake system uses five sorbs with a diameter of 1200 mm, equipped with metallic screens to retain suspended solids [48,49]. The Palas–Constanța water treatment plant (PCTP) provides drinking water to the city’s 350,000 inhabitants through nine large-diameter pipelines. Details on the PCTP can be found in [48].

The geographical locations of the Constanta county and city in Romania are shown in Figure 1.

This study proposes the evaluation of the treatment efficiency of surface water from the Galeșu source and groundwater for the purification and distribution of drinking water to consumers. The experimental results were obtained from the analyses of the surface water and groundwater quality from four sampling points of the treatment plant, denoted by S1–S4 in Figure 2 and representing (1)—raw surface water, (2)—raw pre-chlorinated groundwater, (3)—treated surface water, and (4)—drinking water distributed to consumers from the treatment plant.

The monitored parameters include temperature—T (°C), pH [SR ISO 10523:2012] [50], electrical conductivity—EC (μS/cm) [SR EN 27888:1997] [51], turbidity—TUR (NTU) [SR EN ISO 7027-1:2016] [52], total hardness—TH (^0^dH) [SR ISO 6059:2008] [53], permanganate index—PMI (mg O_2_/L) [SR EN ISO 8467:2001] [54], free residual chlorine (mg/L) [SR EN ISO 7393-2:2018] [55], ${Cl}^{-}$ (chlorides, mg/L) [SR ISO 9297:2001] [56], ${SO}_{4}^{2-}$ (sulphates, mg/L) [Romanian standard: STAS 3069-87] [57], and nutrients—${NH}_{4}^{+}$ (ammonium, mg/L) [SR ISO 7150-1:2001] [58], ${NO}_{2}^{-}$ (nitrites, mg/L) [SR EN 26777:2006] [59], and ${NO}_{3}^{-}$ (nitrates, mg/L) [SR ISO 7890-3:2000] [60]. PMI provides information on the quantity of oxidizable inorganic and organic substances in water. This index is utilized to assess the quality of the freshwater and treated potable waters in the European Union (EU), according to [52].

The data series consists of the monthly average values of the mentioned parameters for 2016–2019. The obtained values were compared with the maximum allowable values (MAVs) from the Romanian legislation [61]. Given that the water temperature may significantly influence the efficiency indexes because there is a high variation between its value at the treatment plant’s entrance and the distribution system’s entrance, we shall not consider it when building the indices.

### 2.2. Statistical Analysis and Efficiency Indicators

The basic statistics of the recorded data series have been calculated, and the histograms and boxplots have been drawn to show the series characteristics and determine the outliers’ existences.

### 2.3. Efficiency Indices

#### 2.3.1. Efficiency Indices of the Treatment Process at a Given Moment *t*

*1. The individual efficiency at the moment t with respect to the k-th water parameter,* ${ef}_{kt}$, is defined by:(1)$${ef}_{kt}=\left[1-{\left(C_{o,t}\right)}_{k}/{\left(C_{in,t}\right)}_{k}\right]\times 100$$
where ${\left(C_{in,t}\right)}_{k}$ and ${\left(C_{o,t}\right)}_{k}$ are the concentrations of the water parameter *k*, in the input and output at a certain point (2, 3, or 4), at the moment *t*.

*2. The mean cumulated efficiency with respect to n water parameters at the moment t, *${MCE}_{t}$ is defined by:(2)$${MCE}_{t}=\frac{1}{n}\sum_{j=1}^{n}{ef}_{jt}$$

*3. The weighted cumulated efficiency with respect to n water parameters at the moment t, *${WCE}_{t}$ is defined by:(3)$${WCE}_{t}=\left(\sum_{j=1}^{n}{ef}_{jt}{\times w}_{j}\right)/\left(\sum_{j=1}^{n}w_{j}\right)$$
where $w_{j}$ is the *j*-th water parameter weight.

#### 2.3.2. Efficiency Indices of the Treatment Process during the Study Period (*T* moments)

*1. The individual average efficiency with respect to the k-th water parameter,* ${AE}_{k},$ is defined by the Formula (4):(4)$${AE}_{k}=\frac{1}{T}\sum_{t=1}^{T}{ef}_{kt\;}=(1-\bar{{\left(C_{o}/C_{in}\right)}_{k}})\times 100.$$
or by ${JAE}_{k},$ whose formula is [62]:(5)$${JAE}_{k}=(1-\bar{C_{o,k}}/\bar{C_{in,k}})\times 100$$
where
(6)$$\bar{{\left(C_{o}/C_{in}\right)}_{k}}=\frac{1}{T}\sum_{t=1}^{T}{\left(C_{0,t}\right)}_{k}/{\left(C_{in,t}\right)}_{k}$$
and $\bar{C_{in,k}}$ and $\bar{C_{o,k}}$ are the averages of the *k*-th parameter concentrations as input and output of a treatment stage during the study period.

*2. The cumulated average efficiency with respect to n water parameter* is defined by one of the formulas:(7)$$\bar{CAE}=\frac{1}{T}\sum_{t=1}^{T}{MCE}_{t}=\frac{1}{n}\sum_{k=1}^{n}{AE}_{k}$$
(8)$$\bar{JAE}=\frac{1}{n}\sum_{k=1}^{n}{JAE}_{k}$$

*3. The weighted cumulated efficiency with respect to n water parameters* is defined by one of the formulas:(9)$$\bar{WCE}=\frac{1}{T}\sum_{t=1}^{T}{WCE}_{t}=\left(\sum_{k=1}^{n}{\bar{AE}}_{k}{\times w}_{k}\right)/\left(\sum_{k=1}^{n}w_{k}\right)$$
(10)$$WJAE=\left(\sum_{k=1}^{n}{{JAE}_{k}\times w}_{k}\right)/\left(\sum_{k=1}^{n}w_{k}\right)$$

The values assigned to the weights are from 1 to 5, considering the harmful potential of some chemicals to human health. The higher the harm potential, the higher the index is. In the present article, we took advantage of the scientific literature findings related to the water quality indices (WQIs) for drinking water. In the manuscript, we used the indices provided in [63] (that gives the most-used weights attached to different water parameters). The weights utilized here are 1 for pH, EC, free residual chlorine, and chlorides, 2 for permanganate index, sulfates, nitrates, and nitrites, and 3 for ammonia.

The highest value of all indices but pH is 100%, corresponding to a perfect working of the treatment plant. Any positive value indicates a certain degree of efficiency. The closer the value is to 100%, the better the station performance. Negative indices indicate the water treatment plant’s incapability to remove certain elements from the water. The lower the indices are, the worse the treatment plant’s performance is. In the case of pH, efficiency around zero means keeping the pH within almost constant limits (6.5–8.5 recommended).

## 3. Results

### 3.1. Results of the Statistical Analysis

Table 1 contains the basic statistics computed for the water parameters analyzed at the sampling points S1–S4.

Most values are inside the admissible limits. Exceptions are some turbidity, ammonium, and nitrite values, whose maxima are in bold in Table 1. The median was also computed, since the range (difference between maximum and minimum) of some series values or their standard deviations are high, indicating a significant dispersion of the values around the mean. Significant differences between mean and median were found for the highly skewed series (TUR at S1 and S3, and ${NO}_{2}^{-}$ at S3, for example).

The free residual chlorine had values of 0 throughout the study for the influent—site 1—because it did not undergo pre-treatment before entering the treatment plant. For the groundwater, which is pre-treated, the chlorine values ranged from 0.43 to 1.28 mg/L, with an average value of 0.85 mg/L. For the treated surface water, the chlorine values ranged from 0.25 to 0.90 mg/L, with an average of 0.50 mg/L, while the concentration of chlorine in the drinking water in the effluent obtained values between 0.40 and 0.64 mg/L, with an average of 0.56 mg/L. Although the allowed values for potable water are between 0.1 and 0.5 mg/L, they must be achieved throughout the entire distribution network. Therefore, even if the values obtained at the treatment plant exceed the MAV, they are accepted to ensure proper water disinfection in the storage tanks and a minimum required chlorine level in the supply pipes.

The histograms and boxplots of some water parameters are shown in Figure 3 and Figure 4. The histograms of the free residual chlorine series recorded at the first two sampling points are symmetric, whereas those for the last sampling points are slightly skewed. A positive skewness is noticed for the concentration series of nitrate at the last sampling point. Various skewness values are determined, indicating that most series are not symmetrically distributed. Kurtosis shows platykurtic distributions for different series, as, for example, pH, EC,$\;{Cl}^{-},$ ${SO}_{4}^{2-}$, and ${NO}_{2}^{-}$ series at S4. The boxplots of pH, turbidity, EC, and ammonia indicate the outliers’ presence (represented by stars).

By comparison, the other series are more homogenous, with only a few outliers. Figure 4c,d,f point out that there are significant variations in the values of the series recorded at different sampling points, especially for ${Cl}^{-},$ PMI, and EC. The existence of high outliers, especially for the TUR series, will significantly decrease the computed performance indices.

### 3.2. Results on the Efficiency Indices

#### 3.2.1. Results on the Efficiency Indices at a Given Moment *t*

The efficiency indices computation was based on the same output—S4—with respect to the input from S1, S2, and S3, respectively denoted by the corresponding indicator followed by S1, S2, and S3. For example, the ${MCE}_{t}\_S1$ means that the input series is from S1. The water temperature and free residual chlorine are not considered in this study because the free residual chlorine is absent in the surface water and groundwater (being added during the purification process), and the temperature does not impact the drinking water quality. Models of free residual concentration series are presented in [48].

The variations in the individual efficiency computed by (1) are represented in Figure 5. Their minimum and maximum values are listed in Table 2.

Some aspects related to individual efficiencies ${ef}_{kt}\_S1$, are presented below:The values of ${ef}_{kt}\_S1$ varied in very large intervals, from negative values for all but PMI and ${NO}_{2}^{-}$ to the maximum (100%).TUR’s efficiency values are all positive, half being 100, except for two negative values (−73.91 and −61.67 in May and June 2017).The maximum individual efficiency of EC is zero, and more than 80% of chloride efficiencies are negative, meaning that the values recorded in the effluent are lower than those in the influent.PMI is the only index whose efficiency values are positive. This means there is good performance in removing the humic materials and organics that could result from the birds and fish exhausts or decomposition.The value of −450 for ammonia is due to a jump from 0.01 (mg/L) in the input to 0.120 (mg/L) in the effluent in August 2019. Another negative value (−200 in November 2019) is noticed in the ammonia efficiency ${ef}_{kt\_S1},$ due to a concentration change from 0.01 to 0.03 mg/L.Negative efficiencies were recorded in June 2017 (−173.53), July, August, October, and November 2021 for ${NO}_{3}^{-}$ and June–October 2019 (less than −137.2) and May–August 2017 (less than −81.79) for ${SO}_{4}^{2-}$.

The analysis of ${ef}_{kt}\_S2$ and ${ef}_{kt}\_S3$ shows that, generally, the maximum efficiencies increased and the minimums decreased. Specific remarks for ${ef}_{kt}\_S2$ are as follows:All values computed based on the pH are positive.The only negative value of efficiency in the TUR series is −250, recorded in March 2016, with half of the values being 100 (excellent efficiency).The lowest negative individual values of efficiencies for EC, ${Cl}^{-}$, and ${SO}_{4}^{2-}$ (under −79.90, −128.46, and −60.32, respectively) are computed for May–December 2017. Moreover, ${Cl}^{-}$ efficiency is mainly negative.Only nine values of the individual efficiency for ammonia are noticed, the lowest being −900, −1200, and −600 (recorded in May–September 2019, May and August 2018).All ${NO}_{2}^{-}$ efficiency indices are positive except six, recorded, for example, in November 2018, and January and December 2019 (with values of −200 and −100).In total, 22 values of ${NO}_{3}^{-}$’s efficiencies are negative, most of the positive ones being under 30.

The lowest individual efficiencies are ${ef}_{kt}\_S3$, as explained in the following:All values corresponding to pH are in the interval [−6.72, 11.66], with most being negative, so an increase in the water’s pH appears when the influent is considered the series at S3 and the effluent is the series at S4.The efficiency of TUR recorded unexpected low values (marked with bold letters) in Table 2, as −2871.43 followed by −1918.20, in June and September 2019, respectively. Most values around −500 were also recorded in February, April, June–November 2018. Some explanation of these values are presented in the next section.The lowest chloride efficiencies were in the range [−155.84, −113.64] and between (−352.01) and (−117.27) in July, August, October, and November 2016, and in June–August and October–December 2017.The lowest values (negative) of the sulfates’ efficiencies were recorded during the same period as those of ${Cl}^{-}$.PMI recorded extremely low efficiencies in the same months as TUR. The value marked in bold was registered in February 2018.

Comparing the period where some values of the individual efficiencies were extremely low, we think that this situation is the consequence of the malfunctioning of the PCTP during May–December 2017.

The series of mean (and weighted) cumulated efficiencies with respect to all water parameters ${MCE}_{t}\;\left({WCE}_{t}\right)$ are represented in Figure 6.

Given that the values −2871.43 and −2118.75 are outliers, we removed them from the computation. Due to the weights assigned to the water parameters as a function of their contribution to the water quality, when most values of the individual indices were positive, ${WCE}_{t}>{MCE}_{t}$. In the case of the negative values, the inequality is the opposite. Comparisons of the extreme values value of both indices are given in Table 3.

The charts from Figure 6 indicate mainly negative cumulated efficiencies computed when the input data series were S3, compared to the case when the input was from S2 or S1, respectively. The best cumulated efficiencies were recorded in the last case—columns 2 and 5 in Table 3. Still, even when using the weighted indices, the cumulated efficiency remains under 50.

#### 3.2.2. Efficiency Indices of the Treatment Process during the Study Period

The individual average efficiencies permit determining the water parameters whose efficiency should be improved considering the recorded values during the entire study period. The efficiencies of the PCTP with respect to each water parameter—(4) and (5)—are presented in Table 4.

Based on ${AE}_{k}$ and ${JAE}_{k}$, the best performances are those of nitrites, turbidity, and ammonia removal with respect to S1. The same parameters remain positive for PMI and TUR (with respect to S1 and S2).

The values of ${JAE}_{k}$ are generally higher than those of ${AE}_{k}$ because the average of input and output series are computed, diminishing the difference between the computed values. Therefore, ${JAE}_{k}\_S3$ has all values but that for chloride greater than zero. Removing the abovementioned, ${AE}_{TUR\_S3}=-$739.52 and ${AE}_{PMI\_S3}=$−381.64 will become 131.62 and −182.46, respectively.

The cumulated average efficiency with respect to the considered water parameters are as follows:
$\bar{CAE}\_S$1 = 5.11, $\bar{CAE}\_S$2 = −8.43, and $\bar{CAE}\_S$3 = −130.1. When eliminating the highest outlier, $\bar{CAE}\_S$3 = −39.84.$\bar{JAE}\_S$1 = 15.61, $\bar{JAE}\_S$2 = 10.74, and $\bar{JAE}\_S$3 = 15.61.

The weighted cumulated efficiency with respect to all parameters are as follows:
$\bar{WCE}$_S1 = 17.61, $\bar{WCE}\_S$2 = −4.69, and $\bar{WCE\_S}$3 = −46.24. When eliminating the highest outlier, $\bar{WCE\_S}$3= −73.10.WJAE_S1 = 30.21, WJAE_S2=21.96, and WJAE_S3 = 17.55.

Considering the cumulated indices that reflect the global efficiency in time and considering all water parameters, we remark a very low performance of the PCTP in time with respect to each input source. The highest one is with respect to the influent from S1.

## 4. Discussion

This article proposes different categories of indices for evaluating the efficiency of water purification of a drinking water treatment plant. These provide a synthetic modality for achieving the goal, given that when working with hundreds of values over a long period it is difficult to look at the charts or the individual values for each day, and it is time consuming. Moreover, determining a model that can be used for forecasting is also difficult in the presence of extreme values or outliers. Computing the indices’ values can be easily accomplished (in an Excel file, for example), and the obtained values (positive or negative, close to 100, for example) will provide a quick answer related to the efficiency of the cleaning process.

The necessity of obtaining individual indices as high as possible for all water parameters but pH comes from the importance of each water parameter, which will be discussed shortly here.

Maintaining the pH for the drinking water between 6.5 and 8.5 is essential given that increased alkalinity of acidity can lead to pipe damage (favoring the detachment of tiny particles from the pipes’ materials) and the impurities’ circulation in the distribution system. Therefore, the water becomes unhealthy for the organism [64,65]. Therefore, negative efficiencies with respect to pH during a long period raise an alarm signal that the pH would be above 8.5, whereas an increasing trend of the individual efficiency with respect to pH will indicate possible pH’s decay under 6.5.

Turbidity indicates that the water is clean from the viewpoint of its aspect (transparent, without suspensions). It is known that water characteristics can undergo significant changes in a short period; for example, turbidity can be strongly affected by heavy rainfall. Increased water turbidity can result from runoff or soil erosion, especially following heavy rains. Therefore, additional operations should be applied at the water treatment plant to prevent hazards and manage associated risks: (a) rainwater storage and management; (b) construction of retention basins to minimize the effects of heavy rains on water quality; (c) advanced water filtration through the use of activated carbon or membranes; (d) addition of coagulants and flocculants; and (e) adjustment of operational parameters [66].

The values of ${ef}_{TURt}\_S1$ indicate that the water was clean from the aspect viewpoint. As for the recorded increased values for water turbidity in May and June 2017, or February, April, June, and November 2018, it is considered that they were mainly determined by abundant precipitation leading to massive water runoff that could accumulate high quantities of soil particles, sand, and other impurities at the treatment plant through the alluvial deposits from flood periods in the area where the treatment plant is located. Other secondary contributions could include construction works, intensive agriculture, or soil erosion.

The individual efficiency indices with respect to EC have mostly negative values, showing an increase in the conductivity values in the output with respect to those in the input. Even if high values of EC do not directly impact health, the large amount of dissolved ionizable solids leads to water hardness and consumer dissatisfaction [67].

PMI is the only water quality indicator with respect to which the PCTC’s individual efficiency values are positive, indicating the correct removal of dead organic material form water.

The very low ${ef}_{kt}\_S3$ (*k* represents the chloride) be explained by the overlap of the station modernization’s works and the seasonal variations in the summer months when higher temperatures and exposure to solar radiation can lead to changes in water composition, including more intense biological activity of organisms (e.g., algae). Additionally, drought episodes leading to decreased water levels through evaporation, altered water composition, or the intensified use of fertilizers in agriculture during the late autumn campaign could contribute to these variations. Another possible explanation at the drinking water treatment plant is the potential infiltration from other nearby sources during maintenance works at the station.

The idea of introducing cumulated efficiency indices was issued from the authors’ previous studies in the water quality indicators field. These indices reflect the water treatment plant’s efficiency with respect to all the considered water parameters. The outliers’ existence in any data series impacts the individual efficiencies and the cumulated ones to a certain extent. A high weight assigned to a parameter with a low (high) individual efficiency will lead to a decrease (increase) in the weighted cumulated efficiency with respect to the average cumulated efficiency. Still, the weighted indices better reflect the impact of each water parameter on the water quality and, consequently, on people’s health.

Given that there are slight variations in the weights assigned by different authors to the same water parameters, the introduced indicators may incorporate some percentage of subjectivity that might be eliminated by averaging the values of the weights found in the literature. Future studies should be performed in this direction.

## 5. Conclusions

This article introduces some indicators used in a case study for assessing the efficiency of a water treatment plant. Whereas the individual indicators show the efficiency with respect to a specific water parameter, emphasizing the issues that may appear on a particular period or with respect to a parameter, the cumulated ones evaluate the overall efficiency over time considering all parameters.

It was shown that the individual efficiencies are sensitive to jumps in values in the effluent with respect to those in the influent (even if they are within the MAV limits). Therefore, the cumulated indices will be drastically affected when very small values participate in their computation. Weighted cumulated indices always differ from the average ones. However, given the importance of each water parameter and the necessity of maintaining good water quality, they must also be observed.

The data analysis indicates that there were periods of malfunctioning of the PCTP, leading to very low negative individual efficiencies with respect to some input sampling points (especially S3) and, consequently, a significant decrement in the cumulated efficiencies concerning each water parameter and the influent. It has been expected that the efficiencies with respect to S3 will be higher given that the water passed through the sedimentation and separation processes. The question that arose was if the maintenance of the water storage tank was correctly performed. To answer this question, sampling should also be performed after exiting the storage tank of 6000 m^3^. A similar sampling should be performed after the storage in the tank of 10,000 m^3^. Unfortunately, at this moment, we do not have such information.

The present study opens the direction for aligning the drinking water quality evaluations with the sustainability objectives (based on objective criteria). Future work will also evaluate the possibility of improving the presented indices and creating a system that will permit the implementation of the necessary corrective measures shortly after they are observed. Moreover, a working methodology must also be determined for the case of outlier existence, given that such values introduce significant biases in the indices’ computation.

## Figures and Tables

**Figure 1 toxics-11-00988-f001:**
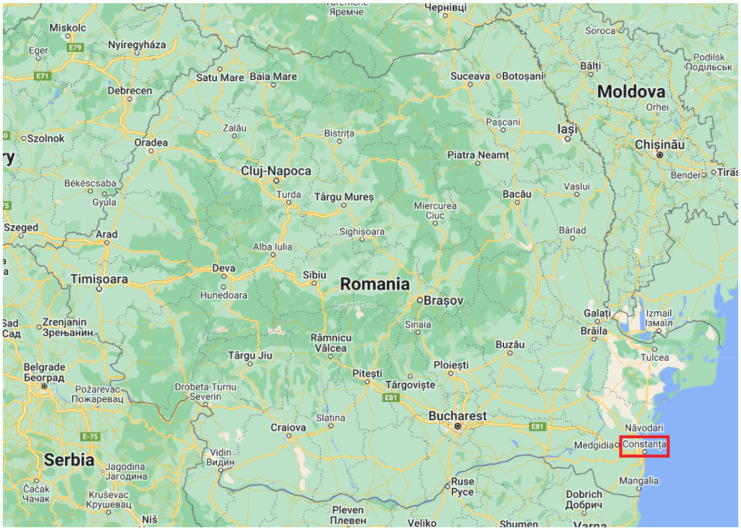
The map of Romania (with Constanța highlighted).

**Figure 2 toxics-11-00988-f002:**
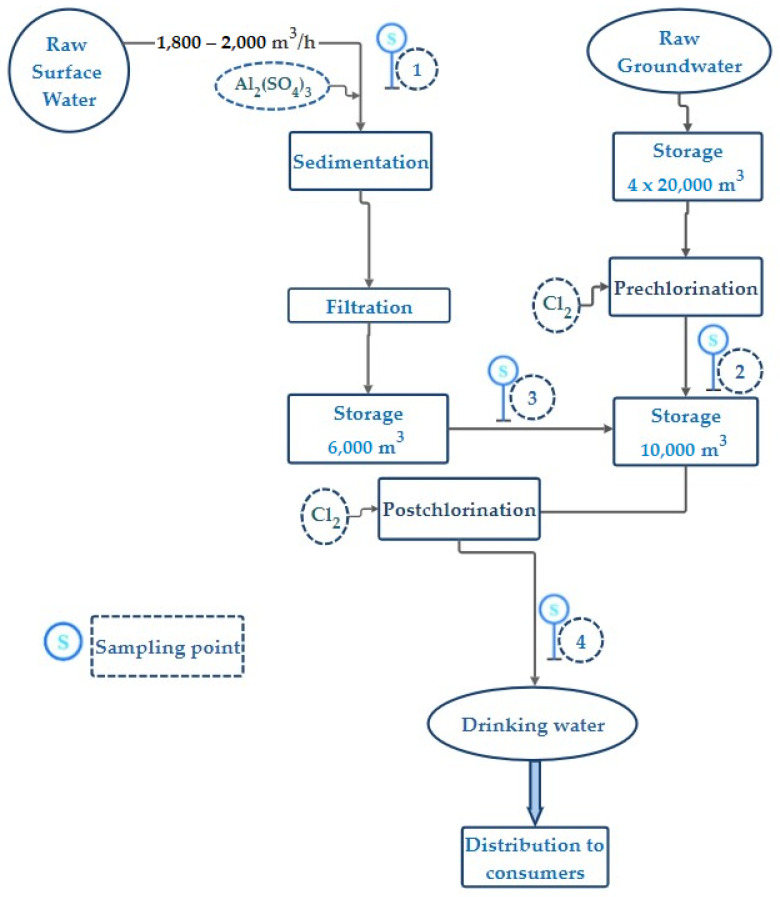
Palas–Constanţa water treatment plant (PCTP) process flow diagram.

**Figure 3 toxics-11-00988-f003:**
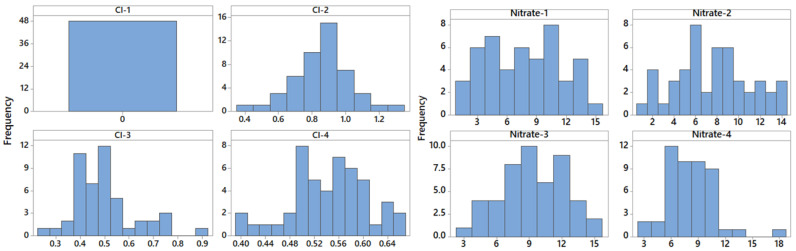
Histograms of free residual chlorine and nitrate at the four sampling points.

**Figure 4 toxics-11-00988-f004:**
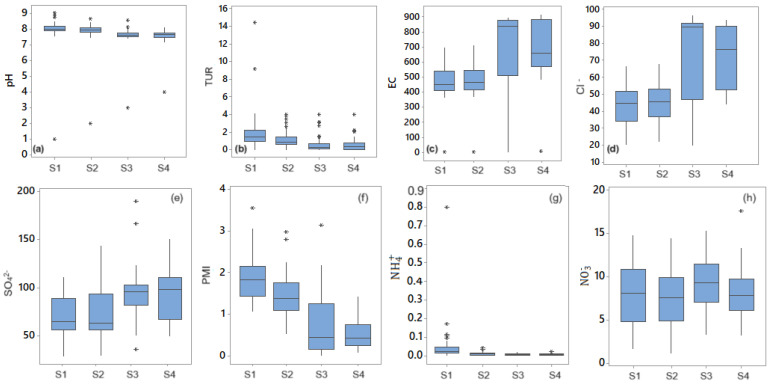
Boxplots of pH, TUR, EC, ${Cl}^{-},$ ${SO}_{4}^{2-}$, PMI, ${NH}_{4}^{+}$ and ${NO}_{3}^{-}$ at the four sampling points (**a**–**h**). Stars represent the outliers.

**Figure 5 toxics-11-00988-f005:**
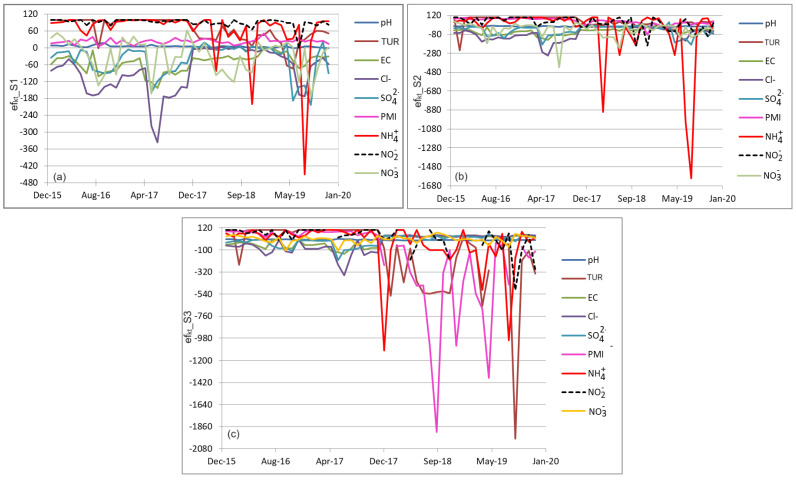
(**a**) ${ef}_{kt}\_S1,\;(b)$ ${ef}_{kt}\_S2,$ and (**c**) ${ef}_{kt}\_S3$. The extreme values (in bold in Table 2) are not represented in the chart.

**Figure 6 toxics-11-00988-f006:**
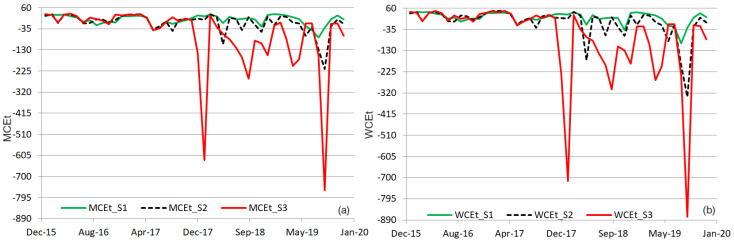
(**a**) ${MCE}_{t}$ and (**b**) ${WCE}_{t}$ computed with the influent from S1, S2, or S3, and the effluent from S4.

**Table 1 toxics-11-00988-t001:** Basic statistics of the water parameters at the sampling points and MAVs [61].

	T (°C)	pH	TUR (NTU)	EC (μS/cm)	${Cl}^{-}$ (mg/L)	${SO}_{4}^{2-}$ (mg/L)	PMI (mg O_2_/L)	${NH}_{4}^{+}$ (mg/L)	${NO}_{2}^{-}$ (mg/L)	${NO}_{3}^{-}$ (mg/L)
Admissible Limit		6.5–8.5	5	2500	250	250	5	0.5	0.10	50
**Sampling point S1**										
min	2.40	7.60	0.00	366.00	20.50	29.60	1.07	0.00	0.00	1.70
mean	14.71	8.11	2.10	479.27	43.37	69.55	1.91	0.05	0.04	7.76
median	15.00	8.03	1.49	450.50	44.45	65.05	1.83	0.023	0.029	8.07
max	26.00	9.07	14.40	696.00	66.40	111.00	3.55	0.80	0.11	14.80
st.dev	7.64	0.29	2.32	86.84	11.35	20.17	0.54	0.12	0.02	3.78
skewness	−0.01	1.31	3.90	0.96	0.01	0.15	0.88	6.11	0.84	0.13
kurtosis	−1.51	2.46	18.16	−0.10	−0.59	−0.78	0.82	39.97	0.33	−1.06
**Sampling point S2**										
min	2.80	7.50	0.00	371.00	22.00	30.40	0.54	0.00	0.00	1.12
mean	15.08	7.95	1.11	486.92	44.94	70.81	1.47	0.01	0.00	7.58
median	14.90	7.96	0.85	466.00	45.65	63.30	1.39	0.008	0.003	7.58
max	26.00	8.69	3.94	712.00	67.40	143.60	2.98	0.04	0.02	14.40
st.dev	7.41	0.26	0.97	85.23	10.73	23.70	0.48	0.01	0.00	3.52
skewness	0.03	0.11	1.50	0.97	−0.03	0.76	0.98	1.56	2.24	0.18
kurtosis	−1.48	0.64	1.83	0.07	−0.46	0.66	1.64	2.60	6.19	−0.73
**Sampling point S3**										
min	3.20	7.43	0.00	371.00	19.80	36.20	0.02	0.00	0.00	3.32
mean	16.61	7.70	0.56	719.46	71.23	92.93	0.71	0.01	0.00	9.37
median	17.15	7.64	0.24	868.50	89.78	96.44	0.45	0.004	0.002	9.52
max	25.60	8.58	4.00	897.00	96.07	189.80	3.14	0.02	0.02	15.30
st.dev	4.87	0.22	0.83	187.54	24.53	26.27	0.68	0.01	0.00	2.96
skewness	−0.68	1.93	2.68	−0.61	−0.53	1.05	1.34	0.99	2.23	−0.05
kurtosis	0.61	4.93	7.64	−1.29	−1.40	3.97	2.05	−0.28	5.62	−0.77
**Sampling point S4**										
min	3.20	7.22	0.00	486.00	44.22	50.60	0.08	0.00	0.00	3.22
mean	16.48	7.65	0.47	718.10	72.30	92.21	0.52	0.00	0.00	8.07
median	17.00	7.68	0.32	670.50	76.43	98.35	0.43	0.003	0.001	7.83
max	24.40	8.12	2.19	915.00	93.50	150.30	1.42	0.02	0.01	17.60
st.dev	4.87	0.22	0.59	151.70	18.52	25.07	0.31	0.00	0.00	2.60
skewness	−0.62	0.21	1.42	0.07	−0.16	0.05	0.77	1.42	0.63	0.94
kurtosis	0.37	−0.67	1.69	−1.82	−1.85	−0.96	0.00	2.38	−0.42	2.62

**Table 2 toxics-11-00988-t002:** The minimum and maximum individual efficiencies.

		pH	TUR	EC	${Cl}^{-}$	${SO}_{4}^{2-}$	PMI	${NH}_{4}^{+}$	${NO}_{2}^{-}$	${NO}_{3}^{-}$
${ef}_{kt}\_S1$	min	−0.51	−73.91	−143.17	−335.61	−202.25	5.41	**−450.00**	0.00	−173.53
	max	17.42	100.00	0.00	3.90	22.45	51.43	100.00	100.00	62.91
${ef}_{kt}\_S2$	min	0.00	−250.00	−139.89	−305.91	−189.23	−82.35	**−1600.00**	**−200.00**	−429.46
	max	12.77	100.00	0.72	6.77	49.86	95.86	100.00	100.00	51.80
${ef}_{kt}\_S3$	min	−6.73	**−** **2871.43**	−139.89	−351.01	−201.10	**−2118.75**	**−1100.00**	**−500.00**	−106.93
	max	11.66	100.00	39.59	50.59	56.76	95.94	100.00	100.00	72.24

**Table 3 toxics-11-00988-t003:** Extreme values of ${MCE}_{t}$ and ${WCE}_{t}$.

	${MCE}_{t}\_S1$	${MCE}_{t}\_S2$	${MCE}_{t}\_S3$	${WCE}_{t}\_S1$	${WCE}_{t}\_S2$	${WCE}_{t}\_S3$
min	−74.27	−215.32	−762.81	−97.33	−336.86	−875.15
max	30.95	31.24	31.97	43.38	47.86	47.82

**Table 4 toxics-11-00988-t004:** Values of (a) ${AE}_{k}$ and (b) ${JAE}_{k}$.

	pH	TUR	EC	${Cl}^{-}$	${SO}_{4}^{2-}$	PMI	${NH}_{4}^{+}$	${NO}_{2}^{-}$	${NO}_{3}^{-}$
${AE}_{k}\_S1$	5.54	66.19	−51.90	−81.21	−42.27	23.56	62.32	91.05	−27.27
${AE}_{k}\_S2$	3.62	55.43	−49.48	−72.51	−41.73	58.63	−37.86	41.09	−33.08
${AE}_{k}\_S3$	0.40	**−739.52**	−12.24	−29.92	−9.21	**−381.64**	−42.53	17.01	5.97
${JAE}_{k}\_S1$	5.63	77.69	−49.83	−66.69	−32.59	23.03	91.51	95.73	−3.97
${JAE}_{k}\_S2$	3.69	57.69	−47.48	−60.87	−30.23	64.86	53.26	62.19	−6.42
${JAE}_{k}\_S3$	0.64	16.81	0.19	−1.50	0.78	27.66	20.91	54.49	13.86

## Data Availability

Data will be available on request from the authors.

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
