# Peer review of "Assessing the Efficiency of a Drinking Water Treatment Plant Using Statistical Methods and Quality Indices"

_toxics, 2023, doi:10.3390/toxics11120988_

Round 1
Reviewer 1 Report
Comments and Suggestions for Authors
The title of the article does not reflect the content. The article concerns the statistical analysis of results calculated using mathematical indicators, and not a direct discussion of the effectiveness of the water treatment plant. Due to the fact that I am not a statistician, it is difficult for me to evaluate the mathematical techniques used to prepare these results. However, what is missing in this text is a precise connection between statistics (mathematics) and the discussion of experimental results in the analyzed period.
What does it mean that the values are positive or negative in the context of the purification technology that is being implemented?
What is the deeper meaning of providing numerical values (example: The efficiency of Tn recorded unexpected low values (marked with bold letters) in Table 2, as -2871.43, followed by -1918.20, in June and Sept 2019, respectively. Most values around -500 were also recorded in Feb, Apr, June-Nov 2018) without interpretation?
PMI – a shortcut to explain, its meaning is not clear to all readers.
The cited literature should be verified >> citation errors
Author Response
Dear Reviewer,
Thank you for the time spent reviewing the article and the valuable comments that helped us improve the article's content. Please find the answers to your comments in the following, highlighted, and the part in red in the new manuscript.
The title of the article does not reflect the content. The article concerns the statistical analysis of results calculated using mathematical indicators, and not a direct discussion of the effectiveness of the water treatment plant. Due to the fact that I am not a statistician, it is difficult for me to evaluate the mathematical techniques used to prepare these results.
We considered your remark and modified the title to reflect the manuscript content better. The new title is “Assessing the efficiency of a drinking water treatment plant by statistical methods and quality indices”
However, what is missing in this text is a precise connection between statistics (mathematics) and the discussion of experimental results in the analyzed period.
We added discussions in the Results and Discussion section. Please see the part in red.
What does it mean that the values are positive or negative in the context of the purification technology that is being implemented?
The highest value of any indices is 100%, corresponding to a perfect working of the treatment plant. Any positive value indicates a certain degree of efficiency. The closer the indice is to 100%, the better the station performance. Negative indices indicate the water treatment plant's incapacity to remove certain elements from the water. The lower the indices are, the worse the treatment plants' performance is.
What is the deeper meaning of providing numerical values (example: The efficiency of Tn recorded unexpected low values (marked with bold letters) in Table 2, as -2871.43, followed by -1918.20, in June and Sept 2019, respectively. Most values around -500 were also recorded in Feb, Apr, June-Nov 2018) without interpretation?
Discussions added. Please see the part in red in the Results and Discussion section.
PMI – a shortcut to explain; its meaning is not clear to all readers.
PMI is the permanganate index. Definition and explanation were added. Please see the text in red in the manuscript.
The cited literature should be verified >> citation errors
You are right. Thank you. We checked the references to avoid citation errors.
Reviewer 2 Report
Comments and Suggestions for Authors
Dear authors,
your paper "Assessing the efficiency of a drinking water treatment plant in Constanta, Romania" needs substantial revision prior to being considered for publication in Toxics.
My general comments are:
1. The "Results" and the "Discussion" parts should be separated and rewritten.
2. The sections of the MS are not correctly numbered there is a repetition of part 2 and parts 3 and 4 are completely missing.
3. The absence of line numbering makes the revision process very difficult. But I will try to explain my
Specific comments:
1. Page 3: Please recalculate and stick to one and the same unit - either m3/h or m3/s.
2. Page 3 and the whole text: Normally turbidity is referred to as "TUR", not "Tn".
3. Page 4: The paragraph "Indices that indicate the efficiency..." should become "2.2.1" instead of "I".
4. Page 5: The paragraph "Indices that show the efficiency of..." should become "2.2.2" instead of "II".
5. Page 5: Format equations 4-10 according to the journal's style, similar to equations 1-3.
6. Page 5: Please add explanations of the symbols used in equations 4 and 5, "where...".
7. Page 5: "Results" should become "3" and "Discussion" should become "4", with their sub-sections also renumbered.
8. Page 6: For the parameters where the StDev is higher than the mean value, the median may be more meaningful to be presented.
Comments on the Quality of English Language
Some corrections should be made. Some sentences are unreadable.
Author Response
Dear Reviewer,
Thank you for the time spent reviewing the article and the valuable comments that helped us improve the article's content. Please find the answers to your comments in the following, in Italics, and the part in red in the new manuscript.
- The "Results" and the "Discussion" parts should be separated and rewritten.
Done.
- The sections of the MS are not correctly numbered there is a repetition of part 2 and parts 3 and 4 are completely missing.
We renumbered the sections.
- The absence of line numbering makes the revision process very difficult.
Sorry for the inconvenience.
But I will try to explain my specific comments:
- Page 3: Please recalculate and stick to one and the same unit - either m3/h or m3/s.
We used now m3/h.
- Page 3 and the whole text: Normally turbidity is referred to as "TUR", not "Tn".
Modified.
- Page 4: The paragraph "Indices that indicate the efficiency..." should become "2.2.1" instead of "I".
Modified.
- Page 5: The paragraph "Indices that show the efficiency of..." should become "2.2.2" instead of "II".
Modified.
- Page 5: Format equations 4-10 according to the journal's style, similar to equations 1-3.
Modified.
- Page 5: Please add explanations of the symbols used in equations 4 and 5, "where...".
Explanations added.
- Page 5: "Results" should become "3" and "Discussion" should become "4", with their sub-sections also renumbered.
Modified.
- Page 6: For the parameters where the StDev is higher than the mean value, the median may be more meaningful to be presented.
The values were added and discussions as well.
Reviewer 3 Report
Comments and Suggestions for Authors
The paper submitted for review concerns the use of contaminant removal indices to characterize the operational efficiency of a water treatment plant in Constanta, Romania. The authors determined individual and cumulative indices based on data characterizing water quality at four sampling points located at different stages of groundwater and surface water treatment technology. In the work, the authors characterized the ground and river water collected by the plant and the method of operation of the plant, and presented the obtained results in detail and transparently along with the designated statistical data. The work is interesting both from the scientific point of view and from the point of view of practical application, but it requires some explanations and supplements.
1. The authors do not explain whether the indexes used have been used before or were developed for this research. It is necessary to present related literature data if they have been used before. If these are new indexes, the genesis of their construction should be presented.
2. It is unclear what new information the individual indices provide compared to the analysis of data on changes in the values of individual parameters characterizing water quality.
3. Please provide the weights assigned to individual parameters when calculating cumulative efficiencies and provide the method of determining the weight values.
4. Information should be provided regarding the basis for interpreting the values of the obtained indices. What values should be considered expected and correct and which indicate incorrect operation of the water treatment plant?
5. There is no discussion of the results obtained in the work, including information on whether these or other indices have already been used in the assessment of the operation of water treatment plants. The operating efficiency of the tested facility should also be compared with data for other stations.
6. The text has been divided into too many paragraphs that do not fit well with the meaning of the information provided.
Author Response
Dear Reviewer,
Thank you for the time spent reviewing the article and the valuable comments that helped us improve the article's content. Please find the answers to your comments in the following, in Italics in red, and the part in red in the new manuscript.
The paper submitted for review concerns the use of contaminant removal indices to characterize the operational efficiency of a water treatment plant in Constanta, Romania. The authors determined individual and cumulative indices based on data characterizing water quality at four sampling points located at different stages of groundwater and surface water treatment technology. In the work, the authors characterized the ground and river water collected by the plant and the method of operation of the plant, and presented the obtained results in detail and transparently along with the designated statistical data. The work is interesting both from the scientific point of view and from the point of view of practical application, but it requires some explanations and supplements.
Thank you for your appreciation.
1. The authors do not explain whether the indexes used have been used before or were developed for this research. It is necessary to present related literature data if they have been used before. If these are new indexes, the genesis of their construction should be presented.
The last paragraph in the Introduction of the initial manuscript answers your question.
“This study aims to fill a gap in the knowledge related to the efficiency evaluation of a water drinking water treatment plant utilizing a series of individual and aggregated indices introduced by the authors. The originality of this work consists of (1) proposing individual and composite efficiency indices for assessing the plant's efficiency, (2) building indices that are not restricted to a certain number of parameters or a determined period, and (3) introducing an objective evaluation method of the treatment plant's efficiency.”
In the new version of the manuscript we added more information.The indices have been proposed by the authors, excepting for JAEk introduced by Juckerski et al. [62] for wastewater treatment (as mentioned in the manuscript). The idea of introducing such indices came after working for some years in the field of evaluation the water quality for drinking purposes quality of the water is using water quality indices (WQI). There are over 200 WQI indices. Since the water treatment for drinking purposes is essential for obtaining good or excellent drinking water quality, we searched for articles related to this topic. They inspired us, and in the article [31] written by the same authors as this article, we introduced a part of the indices used here. We developed our work in this study.
- It is unclear what new information the individual indices provide compared to the analysis of data on changes in the values of individual parameters characterizing water quality.
Thank you for this question, which will help the reader understand the practical value of the proposed indices (if the article is accepted for publication).
Individual indices provide a synthetic modality of assessing the functioning of the water treatment plant. If we work with hundreds of values over a long period – for example, daily data series for ten years – it is difficult to look at the charts or to look at the individual values for each day, and it is time-consuming. Moreover, determining a model that can be used for forecasting is also difficult in the presence of extreme values or outliers. Building the indices can be done very easily, in an Excel file, for example, and the obtained values (positive or negative, closer to 100 or zero, for example) will provide a quick answer related to the efficiency of the cleaning process.
- Please provide the weights assigned to individual parameters when calculating cumulative efficiencies and provide the method of determining the weight values.
The weights are provided in the initial version of the manuscript. They were selected based on the scientific literature related to the WQI computation. In the manuscript, we used the indices provided in the article [63] that present the most used weights attached to different water parameters. These weights are generally assigned on a scale from 1 to 5, considering the harmful potential of some chemicals to human health. The higher the harmful potential, the higher the index is.
- Information should be provided regarding the basis for interpreting the values of the obtained indices. What values should be considered expected and correct and which indicate incorrect operation of the water treatment plant?
Information was added. The positive values indicate correct functioning. The values closer to 100 indicate a very good functioning. Negative values indicate a malfunctioning of the water treatment plant.
- There is no discussion of the results obtained in the work, including information on whether these or other indices have already been used in the assessment of the operation of water treatment plants.
Please see the answer to the first question.
The operating efficiency of the tested facility should also be compared with data for other stations.
Unfortunately, we cannot make comparisons at this point, given that data are not publicly available in Romania.
- The text has been divided into too many paragraphs that do not fit well with the meaning of the information provided.
The number of paragraphs was reduced, but the sections were modified according to the comments of another reviewer.
Round 2
Reviewer 1 Report
Comments and Suggestions for Authors
The authors have significantly improved the article and it may be published in its current form.
Reviewer 2 Report
Comments and Suggestions for Authors
Dear authors,
The paper has been corrected accordingly and could be published in Toxics.
Comments on the Quality of English Languageminor corrections and spell check needed.
Reviewer 3 Report
Comments and Suggestions for Authors
The article has been revised in an acceptable manner and can be accepted for publication in its current form.